# A molecular mechanism underlying gustatory memory trace for an association in the insular cortex

Chinnakkaruppan Adaikkan[1], Kobi Rosenblum[1,2]*

[1]Sagol Department of Neurobiology, University of Haifa, Haifa, Israel; [2]Center for Gene Manipulation in the Brain, University of Haifa, Haifa, Israel

**Abstract** Events separated in time are associatively learned in trace conditioning, recruiting more neuronal circuits and molecular mechanisms than in delay conditioning. However, it remains unknown whether a given sensory memory trace is being maintained as a unitary item to associate. Here, we used conditioned taste aversion learning in the rat model, wherein animals associate a novel taste with visceral nausea, and demonstrate that there are two parallel memory traces of a novel taste: a short-duration robust trace, lasting approximately 3 hr, and a parallel long-duration weak one, lasting up to 8 hr, and dependent on the strong trace for its formation. Moreover, only the early robust trace is maintained by a NMDAR-dependent CaMKII- AMPAR pathway in the insular cortex. These findings suggest that a memory trace undergoes rapid modifications, and that the mechanisms underlying trace associative learning differ when items in the memory are experienced at different time points.

*For correspondence: kobir@psy. haifa.ac.il

**Competing interests:** The authors declare that no competing interests exist

## Introduction

Association between events in time and space is a major mechanism for all animals, including humans, to learn about the world, which can potentially change their behavior in future circumstances. The most influential books concerning learning theory published early in the 20th century were *Animal Intelligence (1911),* wherein Thorndike proposed that the rate of learning diminishes as the interval between response and discomfort/satisfaction is increased (*Thorndike, 1911*), and *Conditioned Reflexes (1927),* wherein Pavlov highlighted that a connection is formed in the nervous system, not between a conditioned stimulus (CS) and unconditioned stimulus (US) separated in time, but between the sensory aftereffects, i.e., trace of the CS, and the US (*Pavlov, 1927*). Thus, in essence, these theories mention that temporal contiguity has a vital influence on associative learning.

The prevailing dogma for cellular mechanisms underlying associative learning is based on modifications of Hebb's pioneering cell assembly theory (1949) which posits that 'cells which fire together, wire together', and thus the firing cells epitomize the internal representations of the two sensory events related in time (*Hebb, 1949*; *Miltner et al., 1999*; *Nabavi et al., 2014*; *Yiu et al., 2014*; *Johansen et al., 2014*). Indeed, it has been shown that at least in certain cases, selective facilitation of densely connected neurocircuits through excitability and/or synaptic plasticity, as a result of an experience, constitutes the basis for learning and memory (*Nabavi et al., 2014*; *Yiu et al., 2014*; *Johansen et al., 2014*).

Most of these studies have examined the cellular and molecular mechanisms underlying trace associative learning processes using classical conditioning paradigms in which the association takes place within the timescale of msec to sec (*Johansen et al., 2014*; *Kitamura et al., 2014*; *Beylin et al., 2001*). However, evolution has produced other type(s) of associative learning, wherein

**eLife digest** The survival of animals, including us humans, depends on the ability to discriminate good food from bad. We would prefer eating a given taste if it did not cause any negative feelings after eating it for the first time; however, we would avoid eating that specific taste if it caused any digestive discomfort. This ability to connect sensory events that happen close in time is called associative learning.

One longstanding theory of associative learning suggests that if the neurons that are activated by a taste fire at the same time as those that control nausea, the connections between the two groups of neurons are strengthened. This helps that particular taste to become associated with the feeling of illness. Animals can also link events that are separated in time – for example, they can become averse to a food even when its ill effects are felt several hours after eating it. An important question is how a new event (such as a new food) is internally represented and maintained for a certain time so that it associates with a response (sickness) that occurs much later.

One method used to investigate associative learning is to feed rats a new food, and then later make them feel nauseous to measure how much this causes them to avoid the food in the future. The gustatory cortex is the part of the brain responsible for perceiving taste.

Chinnakkaruppan and Rosenblum now use this experimental method to investigate the molecular mechanisms in the gustatory cortex that enable the internal representation (or memory trace) of a new taste to be associated with an unwell feeling that occurs much later.

The results of the experiments show that rats will avoid food with a certain flavor if they feel unwell within eight hours of eating it. However, the response of the rats differs depending on when the rat becomes ill. Underpinning these behaviors is the formation of two parallel internal representations of the new taste: a short-term, robust trace that lasts for three hours; and a parallel, longer lasting, weaker trace that lasts for eight hours to associate the taste with its outcome. The weaker, longer-lasting memory trace only forms if the shorter, stronger trace also occurs.

Chinnakkaruppan and Rosenblum found that forming the shorter, stronger memory requires the activity of a signaling pathway in the gustatory cortex that involves biochemical molecules called NMDAR-CaMKII-GluA1. These molecules can increase the strength of signaling between neurons and are already implicated in learning and memory. The next challenge is to put this newly identified molecular mechanism within the relevant neural circuit in the gustatory cortex.

the CS and US can be experienced several minutes to hours apart. For example, the inter-time-interval (ITI) between the CS (novel taste) and the US (visceral information) in the conditioned taste aversion (CTA) paradigm can last for hours (*Garcia et al., 1955*; *1985*; *Kalat and Rozin, 1971*; *1973*; *Chambers, 1990*; *Rosenblum et al., 1993*; *1997*; *Yamamoto et al., 1995*; *Hashikawa et al., 2013*; *Adaikkan and Rosenblum, 2012*; *Stern et al., 2013*; *Inberg et al., 2013*; *Chinnakkaruppan et al., 2014*; *Parkes et al., 2014*; *Sano et al., 2014*). Here, we attempted to understand the mechanisms that enable the taste memory trace to be associated with visceral information after such a long time.

## Results and discussion

In order to evaluate the effect of the ITI between taste and malaise in CTA learning (ITI-CTA) quantitatively and qualitatively, we conducted a behavioral experiment, in which rats were presented with a novel taste (0.1% saccharin; CS) and later were i.p. injected with lithium chloride (0.14 M LiCl; US) at increasing time points ranging between 1 hr and 20 hr (*Figure 1A*). In agreement with previous findings, we found that CTA was acquired after separating the taste and malaise stimuli for up to 8 hr but not 20 hr (*Figure 1B*, *Figure 1—source data 1* and *Figure 1—figure supplement 1*) (*Kalat and Rozin, 1971*; *Kalat and Rozin, 1973*; *Rozin and Kalat, 1971*; *Revusky, 1971*; *Gutiérrez et al., 2003*; *Koh et al., 2009*; *Chinnakkaruppan et al., 2014*). Interestingly, non-parametric cluster analysis revealed that there are two different clusters among 1–8 hr ITI-CTAs: 1–3 hr (hereinafter short-trace) and 4–8 hr (hereinafter long-trace) ITI-CTAs (*Figure 1B*).

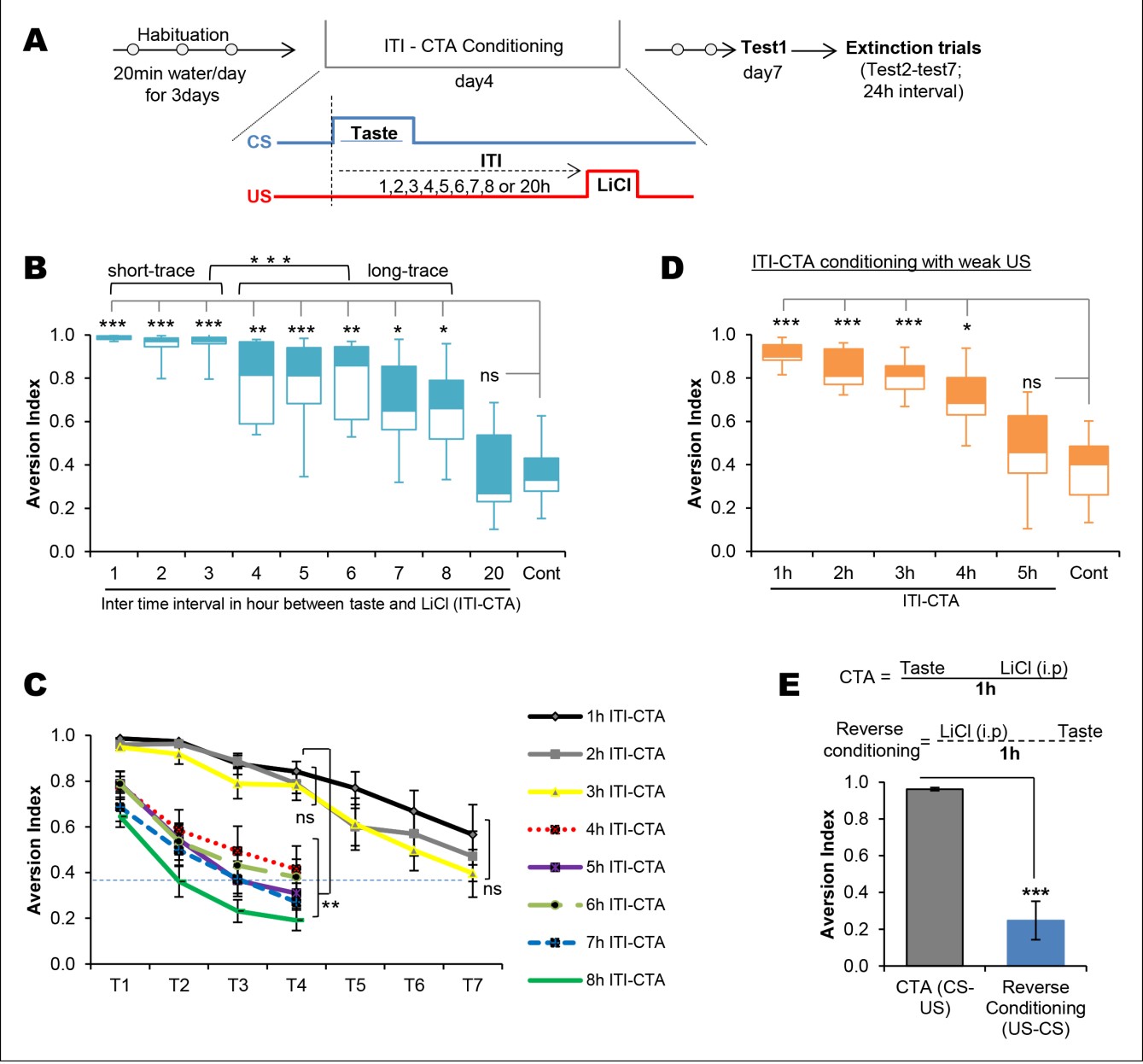

**Figure 1.** Temporal boundaries of taste memory trace for the association with malaise in CTA. (**A**) Schematic diagram of the experimental design. CS, 0.1% saccharin and US, 0.15M LiCl. (**B**) Box-whisker plots showing Test1 results (memory). 1–8 hr ITI-CTA groups but not 20 hr ITI-CTA group showed significantly higher aversion index (AI; a measure of CTA memory) compared to control group. There are two different clusters among 1–8 hr ITI-CTA groups: short-trace (1–3 hr ITI-CTAs) and long-trace (4–8 hr ITI-CTAs). (**C**) 1–3 hr ITI-CTA groups exhibited a similar pattern of extinction and reached AI similar to control group by Test7 whereas 4–8 hr ITI-CTA groups extinguished CTA memory by Test4. Dashed line denotes the mean AI of the control group upon Test1. (**D**) Box-whisker plots showing test1 results following ITI-CTA conditioning with weak US (0.025M LiCl). 1 hr, 2 hr, 3 hr, 4 hr but not 5 hr ITI-CTA groups were significantly different from the control group. (**E**) Top: schematic representation of the CTA conditioning and reverse conditioning trials. In reverse conditioning the novel taste was presented 1 h after the LiCl injection. Bottom: reverse conditioning group did not show CTA. Data in B and D are median and quartile range between 25% to 75%. Data in C and E are mean ± SEM, *p <0.05, **p <0.01, ***p <0.001. n ≥ 5. See also *Figure 1—source data 1* and *Figure 1—figure supplement 1*.

The following source data and figure supplements are available for figure 1:

**Source data 1.** Statistical analysis of *Figure 1*.

**Figure supplement 1.** Attenuation of taste neophobia.

Consistent with the retrieval results, long-trace (4–8 hr) CTA groups exhibited a similar faster CTA extinction pattern, whereas short-trace (1–3 hr) CTAs exhibited a slower extinction. The long-trace CTAs were not different from an unpaired control group by extinction day4, in comparison with day7 aversion index (AI) in short-trace CTAs (*Figure 1C* and *Figure 1—source data 1*). Reverse conditioning did not lead to taste aversion (*Figure 1E*), which indicates the importance of temporal order between taste and nausea in CTA learning. The strength of an association can be affected by ITI between CS and US or by the strength of US. We thus conducted an ITI-CTA with a weak US (0.025M LiCl) to test whether weaker CTA learning can be acquired when an association is made during the short- (1–3 hr) but not the long-trace (4–8 hr) intervals. We found that CTA was formed following the short-trace intervals, and interestingly, also following a 4 hr but not 5 hr trace interval. Together these results suggest that there is a shift in the associability of the taste memory trace at around 3–4 hr after exposure to the taste stimulus (*Figure 1D*).

The long ITI between CS and US in CTA is reasonable from physiological, ecological, and evolutionary perspectives. The minutes to hours process of digestion followed by absorption may dictate the animal to associate between the taste consumed and its delayed physiological consequences (for more information see reviews, *Gal-Ben-Ari and Rosenblum, 2012*; *Kong and Singh, 2008*).

It is possible that depending on the ITI there are two taste memory traces (as indicated by the behavioral readout): the first one is strong, but lasts for approximately 3 hr (short-trace), whereas the second trace is weak, but lasts for up to 8 hr (long-trace). This observation is further supported by the previous demonstration that micro-injection of the protein synthesis inhibitor, anisomycin, into the insular cortex (IC) within 3 hr but not 4 hr after the taste inhibits associative taste memory in the latent inhibition of CTA paradigm (*Merhav and Rosenblum, 2008*). It is therefore plausible to hypothesize that the underlying biological mechanism(s) of taste-nausea association may differ between short- and long-trace CTAs. Next, we set out to test this hypothesis.

Given ample evidence that novel taste experience impacts the phospho-proteome in the IC (see reviews, *Adaikkan and Rosenblum, 2012*; *Gal-Ben-Ari and Rosenblum, 2012*) and the idea that calcium calmodulin-dependent protein kinase II (CaMKII) has the potential to store information (*Lisman, 2014* and the references therein), we aimed to test the possibility that CaMKIIα, through autophosphorylation (at T286; which can act as a molecular positive feedback loop), maintains the taste memory trace in the IC. Moreover, we were encouraged by the previous report that showed that novel taste consumption increases CaMKIIα expression in the IC (*Belelovsky et al., 2005*). Therefore, we followed up this finding and expanded it by measuring the phosphorylation/expression levels of CaMKIIα in the IC for up to 8 hr after the consumption of novel taste. We subjected IC samples of rats exposed to either water or novel taste solution (0.1% saccharin) to biochemical fractionation (*Stern et al., 2013*), and examined the phosphorylation and expression levels of CaMKIIα in the crude synaptosomal fraction (P2-fraction) by Western-blotting analysis (*Figure 2A–C* and *Figure 2—figure supplement 1*). We replicated the finding from the previous report (*Figure 2—figure supplement 2*), that 15 min after novel taste consumption, CaMKIIα expression is increased in the IC. Interestingly, we observed that the T286 phosphorylation of CaMKII (pT286CaMKIIα) was increased in the P2-fraction 30 min following novel taste consumption, persisting for up to 3 hr (1 and 3 hr), but not 5 or 8 hr afterwards (*Figure 2D,E* and *Figure 2—figure supplement 1,2*).

There was no significant difference in T305 phosphorylation of CaMKIIα (pT305CaMKIIα; a phosphorylation site which prevents calcium/calmodulin activation of CaMKII) between water control and novel taste groups at any time point (*Figure 2D–G* and *Figure 2—figure supplement 1,2*). Intriguingly, the temporal dynamics of pT286CaMKIIα in the IC after the novel taste experience corresponds to the short-trace timescale, indicating a possible link between CaMKIIα and short-trace CTA learning.

It is possible that NMDAR activation is upstream to CaMKIIα phosphorylation, because activation of NMDAR in the IC is crucial for taste-malaise association and NMDAR regulates CaMKIIα phosphorylation (*Rosenblum et al., 1997*; *Ferreira et al., 2002*; *Sanhueza et al., 2011*; *Halt et al., 2012*; *Parkes et al., 2014*). Therefore, we examined if NMDAR activation is necessary for novel taste-dependent CaMKIIα activation. Indeed, novel taste experience-induced phosphorylation of T286CaMKIIα is NMDAR-dependent, since i.p. injection of the NMDAR antagonist, MK801 (0.2 mg/kg b.w), 30 min before novel taste learning precluded phosphorylation of T286CaMKIIα in the IC (*Figure 2H,I* and *Figure 2—figure supplement 3*). pT305CaMKIIα and total CaMKIIα levels did not

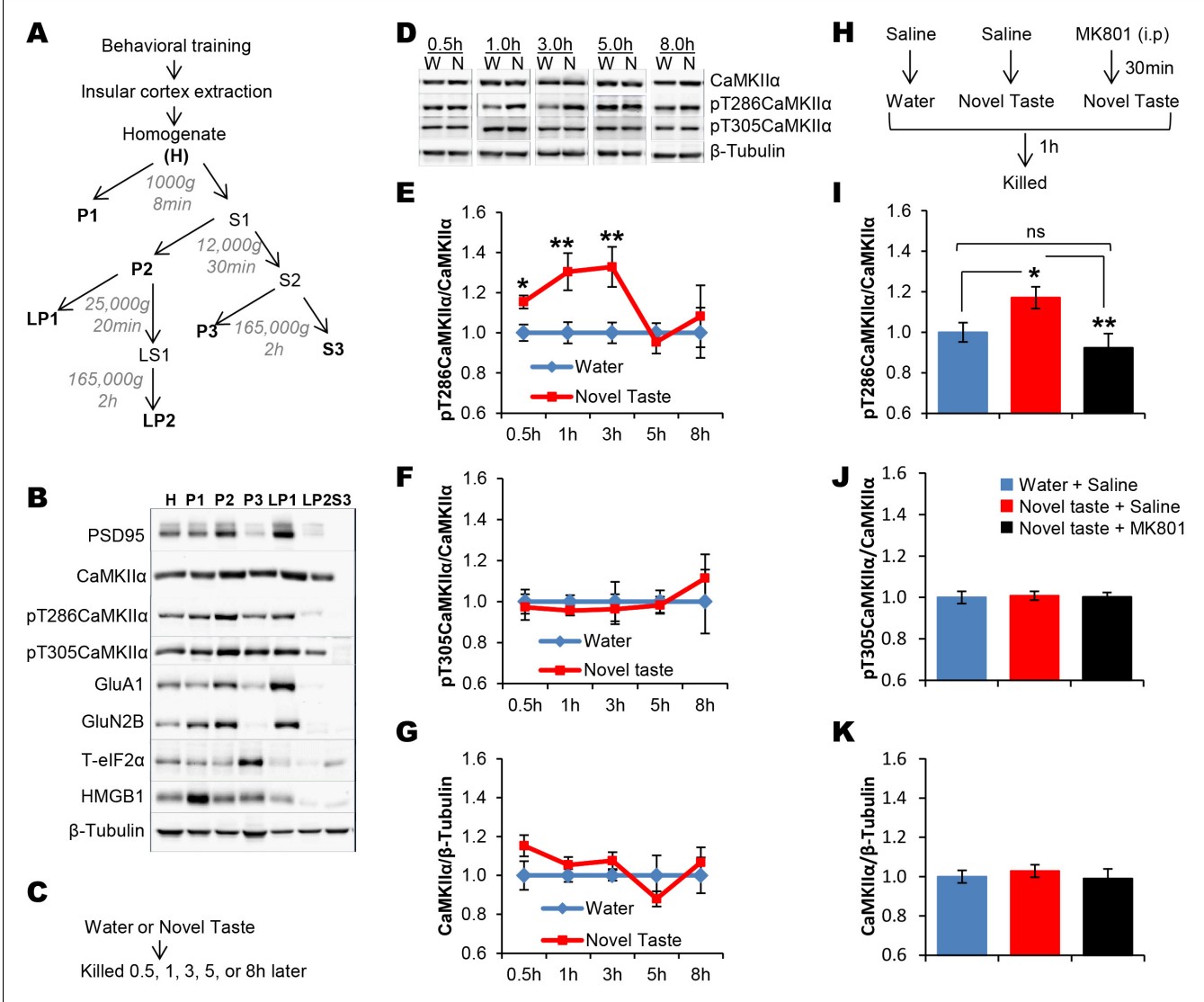

**Figure 2.** Novel taste experience induces CaMKIIα phosphorylation in the IC in an NMDAR-dependent manner. (**A**) Experimental design depicting biochemical fractionation from the IC after behavioral training. (**B**) Representative immunoblots of marker proteins for different fractions. (**C**) Rats were sacrificed at the indicated time points after exposure to either water or novel taste solution. (**D**) Representative pT286CaMKIIα, pT305CaMKIIα, and total CaMKIIα immunoblots from P2-fractions of water (W) and novel taste (N) groups are shown. (**E**) Novel taste groups showed increased pT286CaMKIIα levels in the IC synaptosomal fraction at 0.5, 1, and 3 hr compared to water controls. 5 and 8 hr groups showed no difference in pT286CaMKIIα levels between water and novel taste groups. (**F,G**) There was no difference in (**F**) pT305CaMKIIα and (**G**) total CaMKIIα levels between water and novel taste groups at any time point. (**H**) Schematic representation of experimental design. Rats were injected with saline or MK801 30 min before they were exposed to water or novel taste, and 1 hr later were sacrificed and IC was extracted. (**I**) Novel taste group injected with saline showed increased pT286CaMKIIα compared to water group. MK801-injected groups which received novel taste did not differ from the water group but expressed significantly less phosphorylated CaMKII (pT286) than novel taste group injected with saline. (**J,K**) There was no difference in (**J**) pT305CaMKIIα and (**K**) total CaMKIIα levels between any groups. Data are mean ± SEM, *p<0.05, **p<0.01. n ≥ 6. See also *Figure 2—source data 1* and *Figure 2—figure supplement 1–3*.

The following source data and figure supplements are available for figure 2:

**Source data 1.** Statistical analysis of *Figure 2*.
**Figure supplement 1.** Uncropped immunoblots of main *Figure 2*.
**Figure supplement 2.** Total amount of CaMKIIα in the synaptoneurosomal fractions from IC is increased 15 min following novel taste learning.

*Figure 2. continued on next page*

*Figure 2. Continued*

**Figure supplement 3.** Uncropped immunoblots of main *Figure 2*.

differ among the water group injected with saline and the novel taste groups injected with either saline or MK801 (*Figure 2J,K* and *Figure 2—figure supplement 3*).

Taste memory trace can persist for an association, and at the very same time can undergo sensory information processing to form incidental taste learning (*Chinnakkaruppan et al., 2014*; *Rosenberg et al., 2014*). Therefore, the correlation between the consumption of novel taste and NMDAR-dependent T286CaMKIIα phosphorylation can serve the taste memory trace for an association, and/or incidental taste memory. In order to dissociate between these possibilities, we investigated the role of NMDAR and CaMKIIα in non-associative incidental and associative CTA learning by micro-infusing the NMDAR antagonist APV (10 μg/1 μl/hemisphere) or CaMKIIα inhibitor TatCN21 (0.3 nM/1 μl/ hemisphere (*Buard et al., 2010*)) bilaterally into the IC. Consistent with the literature, our data reveal that NDMAR in the IC is dispensable for incidental taste learning but necessary for CTA learning (*Figure 3—figure supplement 1*) (*Rosenblum et al., 1997*; *Barki-Harrington et al., 2009*; *Parkes et al., 2014*). We also found that CaMKIIα in the IC is dispensable for incidental taste learning but necessary for CTA learning (*Figure 3—figure supplement 1,2*).

Given that NMDAR-dependent T286CaMKIIα phosphorylation in the IC is required specifically for CTA, we sought to test whether NMDAR-CaMKIIα signaling in the IC maintains the taste memory trace for the association with the US by infusing the respective antagonist or inhibitor into the IC at various time points between the CS and US. We micro-injected the NMDAR antagonist into the IC 25 min from the beginning of the taste consumption in 1 hr ITI-CTA conditioning with a weak US and observed an attenuated CTA memory, consistent with a previous report in which a strong US was administered (*Rosenblum et al., 1997*; but also see *Ferreira et al., 2002*) (*Figure 3—figure supplement 3A*). Intriguingly, we did not observe any effect when we made a similar manipulation in the IC 4 hr after the taste consumption in 5 hr ITI-CTA conditioning with a strong US (*Figure 3—figure supplement 3B*; all the remaining experiments were done with a strong US). Next, we micro-injected CaMKIIα inhibitor TatCN21 or Tat control (Tatcont) into the IC 25 min from the beginning of the taste consumption in short-trace 3 hr ITI-CTA conditioning, and observed an attenuated CTA memory (*Figure 3B* and *Figure 3—source data 1*). We micro-injected TatCN21 into the IC 25 min (i.e. short-trace timescale) from the beginning of the taste consumption in the long-trace 5 h ITI-CTA conditioning, and also observed an attenuated CTA memory (*Figure 3C*). However, consistent with the temporal dynamics of CaMKIIα phosphorylation in the IC, CaMKIIα inhibition in the IC 4 h after the taste consumption in the long-trace 5 hr ITI-CTA conditioning had no effect on CTA memory (*Figure 3D*).

These results indicate that novel taste consumption induces NMDAR-dependent T286CaMKIIα phosphorylation in the IC, which is necessary for maintaining the short taste memory trace for the association with visceral information. If this is the case, then inhibiting CaMKII 2 hr after novel taste consumption in 3 hr ITI-CTA should still impede the CS-US association. To answer this, we performed a 3 hr ITI-CTA experiment in which we micro-infused CaMKIIα inhibitor TatCN21 2 hr after the taste consumption, and administered the LiCl 1 hr later. We observed a small non-significant effect (*Figure 3E*). It is thus more likely that T286CaMKIIα phosphorylation in the IC is necessary for the development of the taste memory trace for the association and downstream target/s play a vital role in maintaining the taste memory trace for the CS-US association.

CaMKII-dependent modulation of GluA1-containing AMPA receptors and GluN2B-containing NMDARs has been implicated in different forms of learning, memory and synaptic plasticity, and can induce changes in synaptic strength (*Hayashi et al., 2000*; *Whitlock et al., 2006*; *Sanhueza et al., 2011*). Therefore, we investigated whether pT286CaMKIIα modulates GluA1 and/or GluN2B phosphorylation/expression in the IC after novel taste experience. Indeed, experiencing novel taste increased total GluA1 but not pS831GluA1 or pS1303GluN2B in the synaptosomal fraction in the IC 1 hr later (*Figure 4A* and *Figure 4—figure supplement 1*). Linear regression analysis revealed a positive correlation between pT286CaMKIIα and GluA1 expression in animals sampling novel taste (*Figure 4B* and *Figure 4—figure supplement 1*).

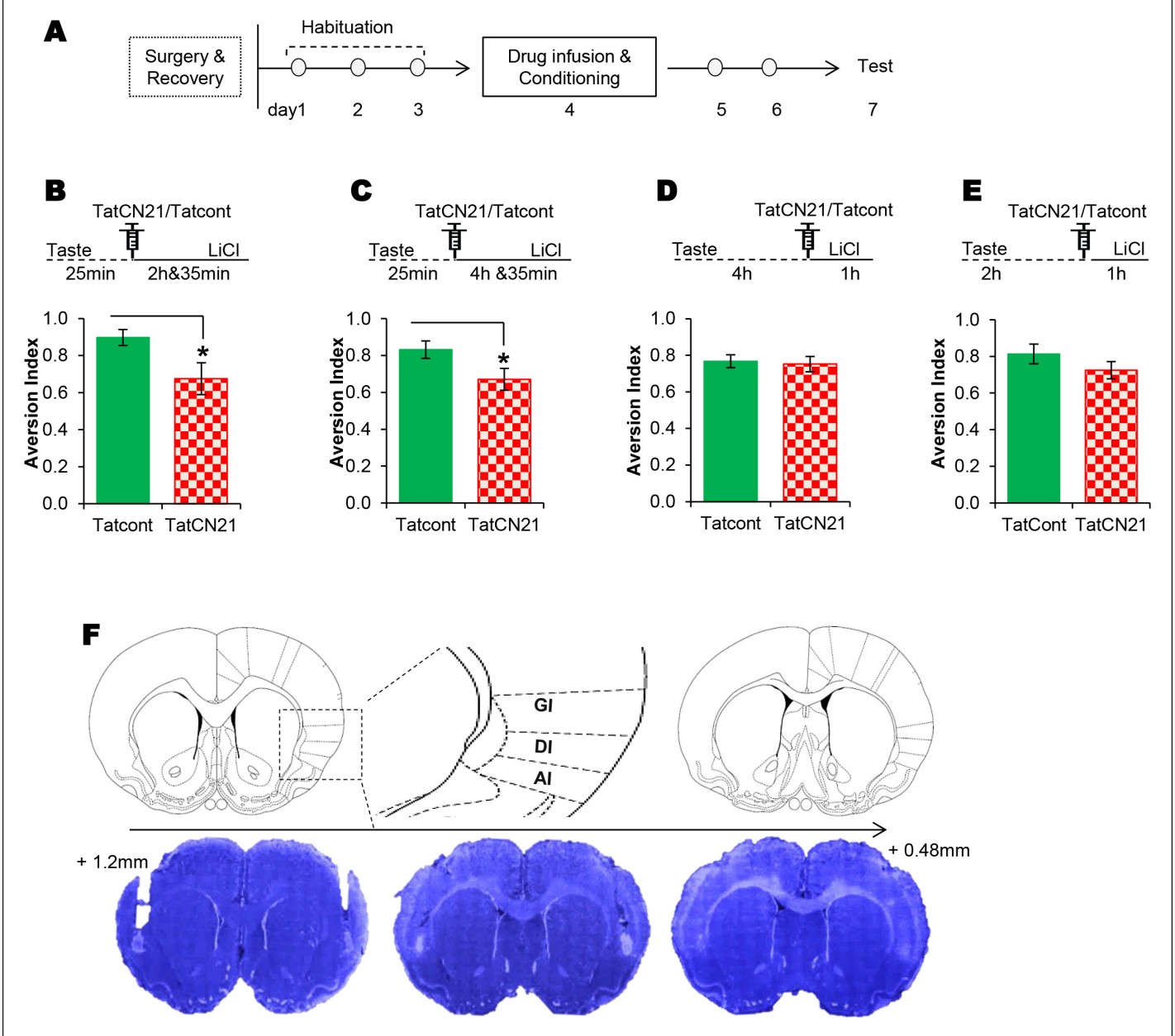

**Figure 3.** The requirement of CaMKIIα in the IC for associative learning of CTA is a function of time. (**A**) Outline of the experimental design. (**B**) Infusion of CaMKIIα inhibitor TatCN21 into the IC 25 min after the taste consumption in 3 hr ITI-CTA conditioning attenuated the CTA memory. (**C**) Infusion of CaMKIIα inhibitor TatCN21 into the IC 25 min after the taste consumption in 5 hr ITI-CTA conditioning attenuated the CTA memory. (**D**) Infusion of CaMKIIα inhibitor TatCN21 into the IC 4 hr after the taste consumption in 5 hr ITI-CTA conditioning had no effect on CTA memory. (**E**) TatCN21 micro-infusion 2 hr after the consumption of taste in 3 hr ITI-CTA training did not affect the CTA memory. Upper panel in B, C, D, and E depict the conditioning trial. The syringe represents microinfusion. (**F**) A series of coronal sections from a representative rat brain, showing the cannula placement in the rostro-caudal planes (lower panel) and the corresponding coronal sections of rat brain atlas images (upper panel). Abbreviations; AI-agranular insular cortex, DI-disgranular insular cortex, GI-granular insular cortex. Data are mean ± SEM, *p <0.05. n ≥ 11. See also *Figure 3—source data 1* and *Figure 3—figure supplement 1–4*.

The following source data and figure supplements are available for figure 3:

**Source data 1.** Statistical analysis of *Figure 3*.

**Figure supplement 1.** NMDAR and CaMKIIα in the IC are required for associative but not for incidental taste learning.

*Figure 3. Continued*

To examine whether pT286CaMKIIα is required for the increased GluA1 expression in the P2-fraction after the taste consumption, we micro-infused CaMKIIα inhibitor TatCN21 into the IC 30 min before novel taste exposure and measured GluA1 expression 1 hr later. TatCN21 application reduced novel taste experience-dependent GluA1 expression in the P2-fraction (*Figure 4C* and *Figure 4—figure supplement 2*). Moreover, NMDAR antagonist MK801 injection 30 min before the novel taste also reduced GluA1 expression (*Figure 4—figure supplement 3*), indicating that NMDAR- and CaMKIIα-dependent increased synaptic expression of GluA1 in the IC mediates short-trace taste memory for the association with the US.

If indeed GluA1 in the IC mediates the taste memory trace for the association, we hypothesized that pharmacological inhibition of AMPAR following taste experience would interfere with the association with the US. Microinjection of AMPAR antagonist, CNQX, (1 μl/hemisphere; 3 nM/μl) (*Tse et al., 2011*) into the IC 1 hr after the taste consumption in short-trace 3 hr ITI-CTA conditioning attenuated the CTA memory (*Figure 4D*). Interestingly, CNQX micro-injection into the IC 2 hr after the taste consumption in 3 hr ITI-CTA learning also attenuated CTA memory (*Figure 4E*). Furthermore, microinjection of CNQX into the IC 1 hr (short-trace timescale) after the taste consumption in long-trace 5 hr ITI-CTA conditioning also attenuated the CTA memory (*Figure 4F*). However, in accordance with the timescale of short-trace and CaMKIIα-GluA1 activation, we found that CNQX micro-injection 4 hr after novel taste consumption in 5 hr ITI-CTA had no effect on CTA (*Figure 4G*).

It is intriguing that the inhibition of CaMKIIα or AMPAR in the IC during short-trace timescale attenuated short-trace CTA memory. In addition, inhibition of the CaMKIIα-GluA1 pathway during short-trace timescale also attenuated long-trace CTA. However, long-trace CTA was not affected when CaMKIIα-GluA1 pathway in the IC was left intact for 3 hr. Together, these data reveal that there are two parallel taste memory traces which are subserved by different mechanisms: one that is robust but decays quickly, and another which is weak but lasts longer.

It is important to note that NMDAR-CaMKIIα-AMPAR in the IC are not necessary for incidental taste learning and memory, whereas muscarinic receptors are critical for incidental taste learning (*Ferreira et al., 2002*; *Parkes et al., 2014*). It is interesting that different molecular mechanisms take place in the same cortex, the insular cortex, to mediate associative and non-associative taste learning and memory.

We should emphasize that different pharmacological manipulation (APV, TatCN21, CNQX) in the IC when applied within 3 hr after the CS presentation only modifies but not completely erases the CTA memory. It is also noteworthy that several previous reports demonstrate that pharmacological perturbations in the IC during CTA conditioning only partially disrupt CTA memory, and that the disrupted CTA appears similar to long-trace CTA memory (*Rosenblum et al., 1993*; *Rosenblum et al., 1997*; *Berman et al., 2000*; *Eisenberg et al., 2003*; *Gutiérrez et al., 2003*; *Barki-Harrington et al., 2009*; *Inberg et al., 2013*; *Stern et al., 2013*; *Parkes et al., 2014*). Thus, it is possible that over time, there is a transformation from the taste memory trace dominance (i.e. short-trace) and IC dependency to multiple memory traces (i.e. during long-trace) due to the multi-channeled background experience, e,g., time, space, food digestion-dependent body physiology, etc. Such transformation, which involves multiple memory traces, for instance, episodic memory trace (how long ago the taste was consumed and under what context?), may result in a wider distribution and processing of the taste memory trace with the passage of time (*Kalat and Rozin, 1971*; *Revusky, 1971*; *Kalat and Rozin, 1973*; *Koh et al., 2009*; *Chinnakkaruppan et al., 2014*).

Since the initial demonstration of CTA by *Garcia (1955),* several studies have proposed that the reduced aversion index followed by the increased interval between the taste and malaise in CTA conditioning could be attributed to (i) the learned safety about the taste, (ii) accumulating background interference, and (iii) taste memory trace decay (*Kalat and Rozin, 1971*; *Rozin and Kalat,*

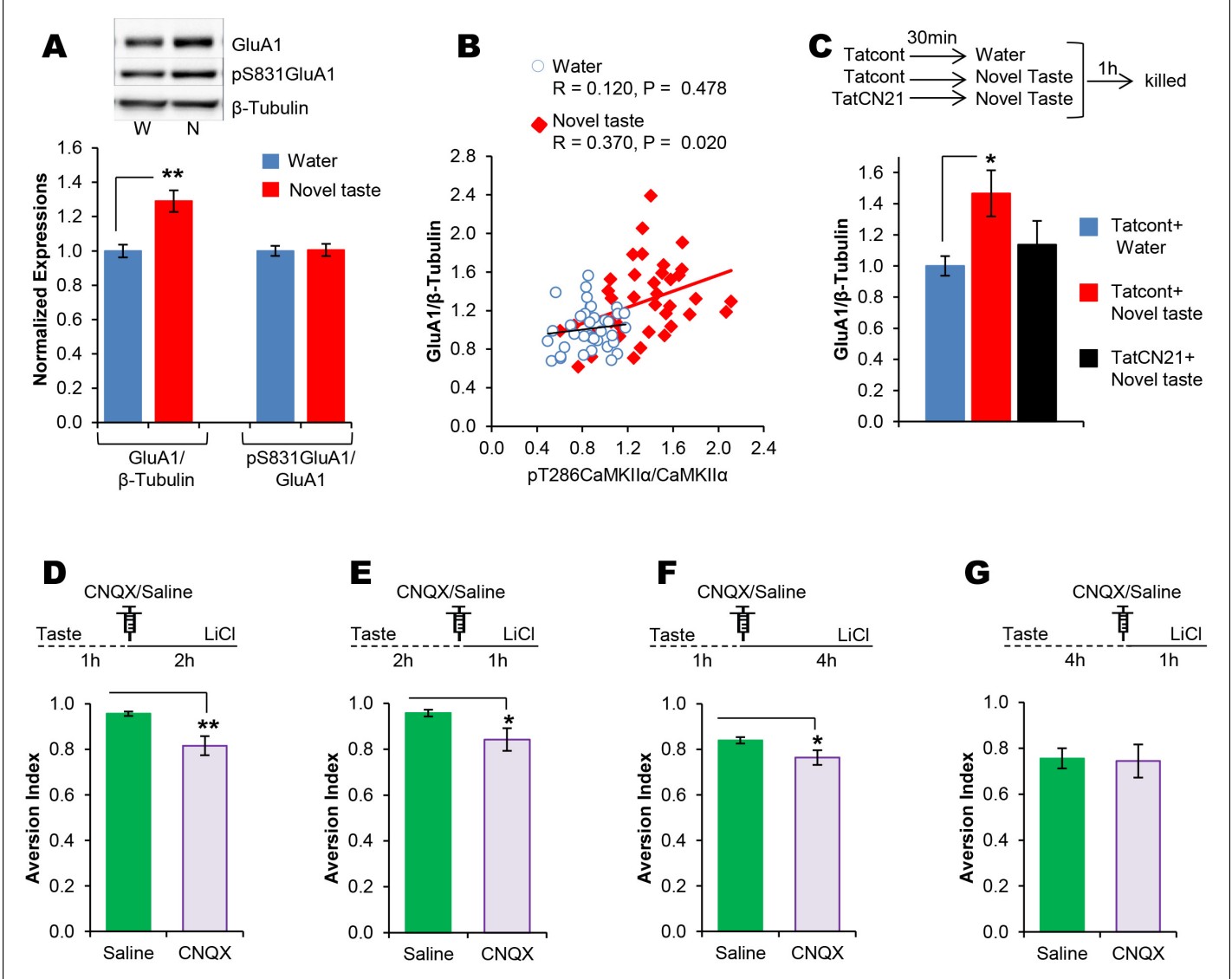

**Figure 4.** The requirement of CaMKIIα-dependent GluA1 expression in the IC for the associative process of CTA is a function of time. (**A**) 1 hr after novel taste consumption total GluA1 but not pS831GluA1 was increased in the P2-fraction. Upper panel shows the representative immunoblots. (**B**) pT286CaMKIIα was positively correlated with GluA1 levels in the novel taste group but not in the water group. (**C**) Upper panel depicts the experimental design. Novel taste-dependent increased GluA1 expression in the IC was precluded by TatCN21 microinjection into the IC. (**D**) CNQX microinjection into the IC 1 hr after the taste consumption in 3 hr ITI-CTA conditioning attenuated the CTA memory. (**E**) CNQX infusion into the IC 2 hr after the consumption of taste in 3 hr ITI-CTA training attenuated CTA memory. (**F**) CNQX microinjection into the IC 1 hr after taste consumption in 5 hr ITI-CTA conditioning attenuated the CTA memory. (**G**) CNQX application 4 hr after taste consumption in 5 hr ITI-CTA conditioning had no effect on CTA memory. Upper panels in D, E, F, and G depict the conditioning trial. Data are mean ± SEM, *p<0.05, **p<0.01. n ≥ 9. See also *Figure 4— source data 1* and *Figure 4—figure supplement 1–5*.

The following source data and figure supplements are available for figure 4:

**Source data 1.** Statistical analysis of *Figure 4*.

**Figure supplement 1.** Novel taste experience leads to an increase in Y1472 but not S1303 phosphorylation of GluN2B.

**Figure supplement 2.** Uncropped original immunoblots of main *Figure 4*

**Figure supplement 3.** Novel taste experience-induced GluA1 in the IC is NMDAR-dependent.

*Figure 4. continued on next page*

*Figure 4. Continued*

**Figure supplement 4.** AMPAR in the IC is dispensable for incidental taste learning.

**Figure supplement 5.** Schematic representation of the conceptual parallel taste-memory trace model.

*1971*; *Revusky, 1971*; *Kalat and Rozin, 1973*; *Gutiérrez et al., 2003*; *Koh et al., 2009*; *Chinnakkaruppan et al., 2014*). First, in support of the learned safety theory, and in line with the previous reports, we found that animals exhibit higher preference to the taste hours following the first-time experience (*Figure 3—figure supplement 4*) (*Kalat and Rozin, 1973*; *Gutiérrez et al., 2003*). Second, with regards to background interference theory, we propose that the multi-channeled background information can be viewed as episodic components and that, for example, the hippocampus is critical to assimilate the episodic component in CTA (*Chinnakkaruppan et al., 2014*; *Koh et al., 2009*). Third, in line with the trace decay theory, our findings demonstrate that the robust short-trace fades within 3 hr in the IC but the long-trace, which is weak, lasts longer. Moreover, the non-linear decay of the short trace suggests that it dominates over the long trace, so that and when the former fades, the latter is revealed (*Figure 4—figure supplement 5*). We propose a multiple memory trace theory, wherein we suggest that CTA learning may engage multiple memory systems to take part in the long-trace associative process and that after experiencing the taste, the IC plays a critical role during short-trace (*Figure 4—figure supplement 5*).

Although multiple lines of evidence support that NMDAR-CaMKII-AMPAR signaling in the IC plays a crucial role in maintaining the short-trace for the association with the US, we do not rule out the possibility that other molecular mechanisms in the IC may participate as well in the long-trace CTA. For instance, protein acetylation, phosphorylation, and induction of proteins were observed in the IC many hours following novel taste consumption, and it is possible that these correlative molecular changes (proteins synthesis and epigenetic regulation) may contribute to the long-trace CTA (*Swank and Sweatt, 2001*; *Yefet et al., 2006*; *Elkobi et al., 2008*; for reviews see *Gal-Ben-Ari and Rosenblum, 2012*).

Conceptually, on the one hand, our short-trace CTA complies with the commonly held assumption of neuroscience theories of associative learning that convergence of CS and US information onto particular cells/circuits leads to changes in synaptic strength at the synapses mediating the CS input to those commonly activated cells/circuit and that it underlies association formation, i.e. Hebbian plasticity (NMDAR-CaMKII-AMPARs in the IC probably in co-ordination with the amygdala; *Yasoshima et al., 2000*; *Ferreira et al., 2005*; *Hashikawa et al., 2013*). On the other hand, the long-trace CTA challenges this assumption, and it is possible that because of the multi-channeled background information with an increasing time interval as discussed above, homeostatic plasticity may play a crucial role to encode multiple memory traces via coordinating several brain structures such as the IC, amygdala, hippocampus, prefrontal cortices in the long-trace CTA (*Vitureira et al., 2012*; *Turrigiano, 2012*; *Pozo and Goda, 2010*). Overall, our data suggest that the neural mechanisms of associative learning can be an assimilation of multiple memory traces when the two relevant experiences are separated in time.

## Materials and methods

### Subjects

The experimental subjects were rats (Rattus Norvegicus; Wistar Hola and Sprague Dawley), obtained from Harlan (Rehovot, Israel). They were maintained at the University of Haifa in a temperature controlled (22–24°C) animal core facility under a 12 hr light/12 hr dark cycle (light phase 7:00–19:00). Experiments were conducted at least 7 days after the acclimatization to the facility when the body weight of the rats was 220-–350 g. Rats were group housed (4–6 rats) in home cages with food and water *ad libitum*, and were individually housed before the start of the experiments. All the experiments were conducted during the light phase.

## Behavioral procedures

### Conditioned taste aversion

Rats were habituated to get their daily water ration (regular drinking tap water) once a day for 20 min from two pipettes, each containing 10ml of water for three days. On the conditioning (fourth) day, they were allowed to drink 0.1% sodium saccharin (Sigma-Aldrich, Rehovot, Israel; prepared in tap water) solution instead of water from similar pipettes for 20 min, and 1, 2, 3, 4, 5, 6, 7, 8, or 20 hr later from the beginning of drinking were injected with lithium chloride (LiCl; Sigma-Aldrich, Israel; 0.15M (for strong CTA) or 0.025M (for weak CTA) prepared in double distilled water; injected i.p. 2% b.w.) for 1, 2, 3, 4, 5, 6, 7, 8, or 20 hr inter-time interval (ITI)-CTA training, respectively. They were given 20 min access to water on days 5 and 6. On day 7 rats were subjected to a more sensitive multiple choice test situation in which two pipettes with 10ml each of saccharin taste solution and two with 10ml each of tap water were presented. The order of the pipettes was counter-balanced and the volume of fluid consumed from each pipette was recorded. The rats were tested again similarly in 24 hr intervals to study memory extinction. The behavioral data are expressed in terms of aversion index, defined as the volume of water consumed divided by the total fluid consumed (water [ml]/water [ml] + taste [ml]).

### Reverse conditioning

Singly housed rats underwent a 3 day water restriction training session in which once a day for 20 min they were offered 20 ml of water from two pipettes, each containing 10 ml. On the conditioning day (fourth day), rats received an i.p injection of LiCl (0.14 M; 2% b.w.) and 1 hr later were offered 20 ml of saccharin for 20 min from 2 pipettes each containing 10 ml. 72 hr post-conditioning the rats were tested similarly as mentioned above and the behavioral data are expressed in terms of aversion Index.

### Incidental taste learning

Singly housed rats underwent a 3 day water restriction training session in which once a day for 20 min they were offered 20 ml of water from two pipettes, each containing 10 ml. On the fourth day, the rats received saccharin solution from 2 pipettes 10 ml each for 20 min. The rats were given 20 min access to water on days 5 and 6 and were tested in a multiple choice test involving two pipettes of water and two of saccharin on day7. The order of the pipettes was counter-balanced and the volume of fluid consumed from each pipette was recorded. The rats were tested again similarly in 24 hr intervals to study the attenuation of taste neophobia. The behavioral data are expressed in terms of aversion index.

### Taste learning and immunoblot analysis

Taste learning and tissue preparation: rats were trained to drink from pipettes for 20 min per day for three days, and on the fourth day they received either water (control) or novel taste (saccharin) for 20 min. They were sacrificed 0.25, 0.5, 1, 3, 5, or 8 hr later. Brains were quickly excised and snap frozen in liquid nitrogen and transferred to -80°C. Insular cortex was extracted from cryostat sections at -20°C. Sub-cellular fractionation was performed according to (*Stern et al., 2013*). Briefly, brain tissues were homogenized (H) in ice-cold homogenization buffer (10 mM Tris-HCl, pH 7.4, 1 mM EDTA, 1 mM EGTA, 320 mM sucrose (all from Sigma-Aldrich, Rehovot, Israel), 1X protease inhibitor mixture (Sigma-Aldrich Rehovot, Israel or Thermo Scientific, Rockford, USA); and 1X phosphatase inhibitor mixture (Sigma-Aldrich or Thermo Scientific). The homogenates were sonicated, kept on ice for 20 min and then centrifuged at 1000 g for 8 min at 4°C to isolate nuclei and large debris (P1). The supernatant (S1) was centrifuged at 12,000 g for 30 min at 4°C in a conventional Eppendorf centrifuge to obtain a crude synaptosomal fraction (P2), and subsequently lysed hypo-osmotically (7 mM HEPES buffer, pH 7.5) and centrifuged at 25,000 g in a Beckman Coulter ultracentrifuge at 4°C for 23 min, to pellet a synaptosomal membrane fraction (LP1). The resulting supernatant (LS1) was centrifuged at 165,000 g for 2 hr at 4°C to obtain a synaptic vesicle enriched fraction (LP2). The supernatant (S2) obtained from the fraction P2 was centrifuged at 165,000 g for 2 hr to obtain a cytosolic fraction (S3) and a light membrane fraction (pellet; P3). Protein quantity was determined with the BCA Protein Assay Kit (GE Healthcare).

Protein Samples were prepared in SDS sample buffer, subjected to 7.5% or 4–20% gradient gel (Bio-Rad pre-cast gels) SDS-PAGE (electrophoresed on Bio-Rad PAGE apparatus) and Western blot analysis. Each sample was loaded with the same amount of total protein (7–10 μg; according to antibody linearity). After transfer to a 0.2 μm pore size nitrocellulose membrane, the blots were blocked with 4% bovine serum albumin (BSA) in tris-buffered saline plus 0.5% tween-20 (TBST) at room temperature for 1 hr. They were then incubated overnight with the suitable primary antibodies: CaMKIIα (1:25,000; Santa Cruz Biotechnology, SCBT), pT286CaMKIIα (1:1000; SCBT), pT305CaMKIIα (1:1000; Novus biological), GluN2B (1:1000; SCBT), GluA1 (1:1000; Abcam or SCBT), pS831GluA1 (1:5000; Epitomics), PSD-95 (1:1000; SCBT), HMGB1 (1:10,000; Abcam), eIF2α (1:1000; Cell Signaling Technology), pY1472GluN2B (1:1000; a gift from Prof. Nakazawa), pS1303GluN2B (1:10,000; Abcam), β-actin (1:3000; SCBT) and β-tubulin (1:30,000; Sigma). The blots were then subjected to three 5 min washing steps in TBST, after which they were incubated with the corresponding HRP-conjugated secondary antibodies: goat anti-rabbit (IgG), goat anti-mouse (IgG) or rabbit anti-goat (IgG) (1:10,000; Millipore Bioscience Research Reagents) for 1 hr at room temperature followed by three 10 min washing steps with TBST. Immunodetection was performed with the enhanced-chemiluminescence EZ-ECL kit (Biological Industries, Israel). The immunoblots were quantified with a CCD camera and Quantity One software (Bio-Rad). Each immunoblot was measured relative to the background and normalized to the endogenous controls (β-tubulin). Phosphorylation levels were calculated as the ratio between the readings from the antibody directed against the phosphoproteins and those from the antibody directed against the phosphorylation state-independent forms of the proteins.

## Pharmacology

### Surgery

The rats were anesthetized with equithesin (2.12% [w/v] MgSO$_4$, 10% [v/v] ethanol, 39.1% [v/v] 1,2-propranolol, 0.98% [w/v] sodium pentobarbital, and 4.2% [w/v] chloral hydrate) at 0.3 ml per 100 g body weight (*Stern et al., 2013*; *Merhav and Rosenblum, 2008*). They were then restrained in a stereotactic apparatus (Stoelting) and implanted bilaterally with $10 \pm 0.02$ mm, 23gauge stainless steel guide cannulae above the IC. Coordinates with reference to Bregma were: A/P = +1.2 mm, L/M = $\pm$ 5.3 mm, D/V = -5.2 mm. The cannulae were fixed in position with acrylic dental cement and secured with two skull screws. Animals were allowed 5–7 days to recover from the surgery before undergoing the experimental manipulations.

### Microinjection

Rats were habituated and the quality of the guiding cannula was checked 24 hr before the microinjections. Rats were brought to the micro-injection room 5–10 min before the microinjection and after careful preparation a 28-gauge, $11.2 \pm 0.02$ mm long injection cannula, extending 1.2 mm from the tip of the guide cannula, was carefully inserted. The injection cannula was connected via PE20 tubing (backfilled with saline) to a Hamilton micro-syringe, driven by a microinjection pump (Harvard) that operated at a rate of 1 μl/min for 1 min. Following injection, the injection cannula was left in place for an additional 1 min before withdrawal to minimize dragging of injected liquid back along the injection track.

### Drugs

Stock TatCN21 and Tatcontrol peptides (a kind gift from Dr. Ulrich Bayer, Department of Pharmacology, University of Colorado School of Medicine, Aurora, Colorado, USA) were prepared in double distilled water to a concentration of 5 mM, aliquoted, and stored at -20˚C. Before the microinjection TatCN21 and Tatcont were further diluted in saline to a final concentration of 0.3 nM/μl and injected. CNQX disodium salt (6-cyano-7-nitroquinoxaline-2,3-dione disodium, Tocris, Bristol, UK) was prepared in saline to a concentration of 3 mM, aliquoted and stored in -20˚C. CNQX was freshly prepared for most of the experiments or was used within 2 weeks of preparation. Before micro-injection CNQX was thawed at 40˚C and injected. APV (2R-amino-5-phosphonovaleric acid, Sigma-Aldrich) was prepared in saline to a final concentration of 10 μg/μl, aliquoted, and stored at -20˚C. Before microinjection APV was thawed at room temperature and injected.

## Histology

At the end of the behavioral experiments, the rats were sacrificed. The brains were removed, frozen with dry ice and kept at -80°C. Randomly chosen rats were micro-injected into the IC with 0.5 or 1 µl/min cresyl violet 30 min before sacrifice. Coronal 50-µm sections were cut in a cryostat, stained with cresyl violet, and examined with a computerized Olympus microscope camera to verify cannula placement.

## Statistical analysis

All grouped data are presented as mean ± SEM. Comparisons between data of two independent groups were analyzed by unpaired Student's t test and the differences between the variances of groups were corrected following Levene's test for equality of variances. Multiple group comparisons were assessed using one way analysis of variance (ANOVA) and repeated-measures ANOVA. Follow-up analyses were conducted using Fisher's least significant difference, Bonferoni, and independent sample t-tests, when significant main effects or interactions were detected. Non-parametric Kruskal-Wallis test, Friedman's test, and Mann Whitney U test were conducted when the data were non-normally distributed. Pearson's correlation was conducted to analyze the association between GluA1 and pT286CaMKIIα. All of the comparisons were conducted using two-tailed tests of significance. The null hypothesis was rejected at the $p < 0.05$ level. Data analysis was performed using SPSS-version 19.

## Acknowledgements

We thank Dr. Ulrich Bayer for providing us TatCN21 and Tatcont peptides, Dr. Takanobu Nakazawa for pY1472GluN2B antibody, and Dr. Thomas McHugh, Dr. Genela Morris, and members of KR lab for helpful discussions. This work was supported by the Israel Science Foundation (1003/12), Morasha (Israeli Science Foundation Legacy Heritage 1315/09), and German-Israeli Foundation DIP (RO3971/1-1) for KR.

## Additional information

### Funding

| Funder | Grant reference number | Author |
| --- | --- | --- |
| Israeli Science Foundation Legacy Heritage | 1315/09 | Kobi Rosenblum |
| German-Israeli Foundation for Scientific Research and Development | RO3971/1-1 | Kobi Rosenblum |
| Israel Science Foundation | 1003/12 | Kobi Rosenblum |

The funders had no role in study design, data collection and interpretation, or the decision to submit the work for publication.

### Author contributions

CA, Conception and design, Acquisition of data, Analysis and interpretation of data, Drafting or revising the article; KR, Conception and design, Analysis and interpretation of data, Drafting or revising the article

### Ethics

Animal experimentation: The procedures were approved by the University of Haifa ethics committee for animal research and were in accordance with the NIH guidelines for the ethical treatment of animals. All of the animals were handled according to the Haifa University animal care and use committee. All surgery was preformed under sodium pentobarbital anesthesia, and every effort was made to minimize suffering.

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
