## [Decision Letter]

Thank you for choosing to send your work entitled "A Molecular Mechanism Underlying Gustatory Memory Trace for an Association in the Insular Cortex" for consideration at *eLife*. Your full submission has been evaluated by a Senior Editor and two peer reviewers, one of whom is a member of our Board of Reviewing Editors, and the decision was reached after discussions between the reviewers. Based on our discussions and the individual reviews below, we regret to inform you that your work will not be considered further for publication in *eLife*.

After careful discussion among the reviewers, we found that your paper contained interesting findings, but the evidence for some key conclusions is too thin and requires a lot more work. We also felt that the rational for the two parallel memory traces are not clear. We therefore came to the conclusion that the manuscript is not suitable for *eLife* at this time. We hope that the comments of the reviewers are helpful in further improving the manuscript.

*Reviewer #1:* The authors investigate a special form of memory, conditioned taste aversion (CTA) which can be often learned in a single trial. They first describe the temporal dependence of convergence of CS and US information for successful establishment of CTA and discover two components with different dwell time and strength, the first lasting app. 3 hours, with robust retentions, while US applied later than 3 hours and until 8 hours showed significant less learning. Subsequent combination of behavioral testing with biochemical analysis showed that after novel taste experience, CaMK2 phosphorylation is increased for up to 3 hours that CaMKII activity is required in IC and that GluA1 receptor number increases.

A major claim of the authors is that the drop in learning efficacy after 3 hours signifies the presence of two parallel mechanisms of CTA acquisition, a claim that is also supported by a previous study from the same group (Merhav & Kornblum, 2008) that shows that protein synthesis inhibitors have a stronger impact on CTA shorter than 3 hours. However, this claim receives overall weak support, as retention of the CS signal may simply be decaying nonlinear with time; or alternatively, the underlying processes linked to novel taste acquisition and induction of CTA crosses below a threshold for efficient learning acquisition after app. 3 hours.

The authors state that short-memory trace is required to generate the weak memory trace which lasts longer, yet the authors claim that the two memory traces are parallel. This is confusing, as when one trace requires the other one, they are by definition not parallel processes.

There are also several issues about the experimental design, in particular appropriate control experiments. For example, the authors show that MK-801 pre injection obliterated the CS impact, and the authors claim this to be a sign that NMDA receptors are required for CS induced CTA. Unfortunately, the result does not allow to distinguish whether NMDA receptors are required for learning or whether it is required for sensory perception. MK-801 as a general blocker of NMDA receptor function may affect many other aspects of CS learning including perception of taste, general sensory processing. Similar concerns relate to manipulations used in Figure 4 using CNQX.

*Reviewer #2:* In this paper, Chinnakkaruppan & Rosenblum used behavioral, biochemical and pharmacological approaches to characterize two different, and potentially parallel, taste memory traces: a short-trace taste conditioning (1-3 h between taste and malaise) leading to strong conditioned taste aversion (CTA) and a long-trace taste conditioning (4-8 h between taste and malaise) leading to weak CTA. In particular, the authors provide an original demonstration that the short-trace CTA critically dependent on CaMKII-GluA1 pathway in the insular cortex whereas the long-trace CTA seems to be partially independent of these processes. The authors then propose that there are two parallel taste memory traces, a stronger one dependent on insular cortex and another weaker one independent of the insular cortex.

In my opinion, the rationale for the study is clearly presented, the experiments are well designed, and the data are appropriately analyzed. I have some major comments meant to encourage the authors to clarify their interpretations of the two parallel memory traces. If these concerns can be addressed the work could yield an important paper.

I encourage the authors to elaborate on the existence of the two parallel memory traces. In particular I suggest several points:

1) Short-trace: in order to clearly demonstrate that the 1-3 h taste trace is dependent on CaMKII-GluA1 pathway in insular cortex, the authors should evaluate the impact of intra-insular infusion of TatCN21 and CNQX 2h after saccharin consumption and 1h before LiCl (ITI 3h). If their assumption is right these treatments should still impair 3h ITI-CTA.

2) Long-trace: the authors clearly demonstrate that the long-trace (> 3h ITI) is not dependent on CaMKII-GluA1 pathway in insular cortex. However, as they have previously shown that novel taste is able to enhance protein expression (C/EBPbeta) in insular cortex long after its consumption (i.e. 18h, Merhav et al. 2006), they should consider that the long-trace CTA could still depend on molecular mechanism in insular cortex (but not on CaMKII-GluA1 pathway).

3) Long-trace: the authors propose that the taste memory trace results in a wider distribution with the passage of time and that there is competition across multiples memory systems. Therefore they should propose a network of brain areas able to sustain the long-trace different from the network sustaining short-trace.

[Editors’ note: the authors appealed against the decision and were later encouraged to resubmit.]

Thank you for choosing to send your work entitled "A Molecular Mechanism Underlying Gustatory Memory Trace for an Association in the Insular Cortex" for consideration at *eLife*. Your submission has now been considered by a Senior editor, the original Reviewing editor, and an additional reviewer, and we are prepared to consider a revised submission with no guarantees of acceptance. The third reviewer's full comments are included below. Before you submit your revised paper please include a thorough explanation of the model and provide additional data.

*Reviewer #3:*

In this article, Chinnakkaruppan & Rosenblum raise an interesting question regarding the nature of memory trace formation following conditioned taste aversion, a special form of single trial associative learning where rats can acquire a memory for a specific taste (CS) associated with malaise (US) up to several hours following the presentation of the CS. They suggest the possible existence of two parallel taste memory traces: one which is strong but limited in time and lasts up to 3 hours and a second one weaker but longer in duration and lasts up to 8 hours from CS presentation. Furthermore, by applying biochemical and pharmacological techniques they suggest the involvement of specific molecular mechanism mediated by CamKII-GluA1 pathway that underlies the strong and shorter memory type but not of weak and longer memory type. Importantly, their results help to explain previous results from the same and other groups regarding the partial disruption of CTA memory by some pharmacological interventions. The possibility of parallel memory traces, which are dependent on separate molecular mechanisms, is exciting. The presented data is highly interesting with a clear potential to make it into *eLife*.

However, I have several concerns regarding the conclusions of the authors about the two parallel memory traces, which should be addressed before the article can be accepted for publication.

1) The authors should at the very least hypothesize regarding the possible molecular mechanisms that might underlie the long but weak memory trace formation (e.g. NMDAR dependent), or even better test their hypothesis by applying the relevant pharmacological intervention.

2) Related to this issue, it would be important to examine the effect of intra-insular infusion of MK-801 at a time interval longer than 3 hours to examine the possible effect of NMDAR on the weak memory trace.

3) Although the authors nicely demonstrate the temporal effects of the CTA (CS + US) acquisition traces, there is no quantification of the impact of the novel taste (CS) and malaise (US) on the formation of these memory traces. The authors could apply a weak CTA learning by reducing the LiCl concentration to address this question. Alternatively, they could induce weak learning by using latent inhibition, where the CS is familiarized before association. They should test: (a) whether this weaker learning can be acquired when association is made during the short (1-3 hrs) interval and long (4-8 hrs) interval and (b) is it sensitive to the CaMKII-GluA1 proposed mechanism or perhaps to a NMDAR dependent one.

---

## [Author Response]

Reviewer 2 was extremely positive towards our manuscript and made some excellent suggestions on experimental and editorial additions that we are happy to comply with. Given this, we were surprised that the manuscript was rejected based on the opinion of Reviewer 1. As detailed in our point-by-point response below, data which address most, if not all, of the concerns that Reviewer 1 raised was already present in the supplemental data portion of the submitted paper. We apologize if this was not made clear in our initial submission and are happy to address these semantic shortcomings in the revised version.

Reviewer #1:

*[…] A major claim of the authors is that the drop in learning efficacy after 3 hours signifies the presence of two parallel mechanisms of CTA acquisition, a claim that is also supported by a previous study from the same group (Merhav & Kornblum, 2008) that shows that protein synthesis inhibitors have a stronger impact on CTA shorter than 3 hours. However, this claim receives overall weak support, as retention of the CS signal may simply be decaying nonlinear with time; or alternatively, the underlying processes linked to novel taste acquisition and induction of CTA crosses below a threshold for efficient learning acquisition after app. 3 hours.* We would like to emphasize that three different lines of evidence (behavioural, Figure 1; biochemical, Figure 2 and Figure 4; and pharmacological, Figure 3 and Figure 4) support the possibility of two parallel taste memory traces.

The reviewer commented that ‘retention of the CS signal may simply be decaying nonlinear with time’. It is clearly a possibility; however, our experiments and the respective statistical analyses did not support this possibility, but instead support the parallel memory traces hypothesis.

First, non-parametric one way ANOVA (Kruskal-Wallis) and non-parametric repeated measures ANOVA (Friedman’s) revealed that the aversion index measure following 1h, 2h, and 3h ITI-CTAs significantly differs from that of 4h-8h ITI-CTAs (Figure 1 and [Supplementary-material SD1-data]). The follow-up unbiased cluster analysis revealed that 1h-8h ITI-CTAs fall into two different clusters; 1h,2h, and 3h ITI-CTAs as small cluster, and 4h-8h ITI-CTAs as large cluster and they are statistically different from each other (Mann Whitney U test; [Supplementary-material SD1-data]). Moreover, as we have presented in the revised manuscript (Figure 1), the associability of the CS with the weak US completely drops at around 4h-5h.

Secondly, in striking opposite to the reviewer’s comment, in Figure 3—figure supplement 4 we have demonstrated (consistent with the previous reports, Kalat and Rozin, 1973; Gutierrez et al., 2003) that the retention of the CS signal gets stronger with time.

The reviewer again presented an alternative comment ‘the underlying processes linked to novel taste acquisition and induction of CTA crosses below a threshold for efficient learning acquisition after app. 3 hours’. This is counter-intuitive to what we have presented in the manuscript: that the underlying process (NMDAR-CaMKII-GluA1 pathway) for formation of CTA with upto a 3h ITI is different from those who have an ITI >3h (4h-8h ITI CTA).

*The authors state that short-memory trace is required to generate the weak memory trace which lasts longer, yet the authors claim that the two memory traces are parallel. This is confusing, as when one trace requires the other one, they are by definition not parallel processes.*

We see the point of the reviewer. We would like to clarify this point with a graphical representation (Figure 5).

Author response image 1.(**A**) Immediately after ingesting a novel taste a robust taste memory trace is generated and it lasts for about 3h. (**B**) concurrently, the weak taste memory trace is also generated and it lasts longer, for about 8h. (**C**) As presented in the manuscript that “inhibition of the CaMKII-GluA1 pathway during short-trace timescale attenuated long-trace CTA, however, long-trace CTA is not affected when CaMKII-GluA1 pathway in the IC was left intact for 3h” and that we suggest that there are two parallel taste memory traces; one that is robust but decays quickly and the second which is weak but lasts longer.**DOI:**
http://dx.doi.org/10.7554/eLife.07582.024

Furthermore, as we mentioned in the submitted manuscript “that several reports including ours demonstrate that pharmacological perturbations (before the acquisition or during the short-trace timescale) in the IC during CTA conditioning only partially disrupt CTA memory, and that the disrupted CTA appears similar to long-trace CTA memory (Rosenblum et al., 1993; Rosenblum et al., 1997; Berman et al., 2000; Eisenberg et al., 2003; Gutiérrez et al., 2003; Barki-Harrington et al., 2009; Inberg et al., 2013; Stern et al., 2013; Parkes et al., 2014)”. Therefore, we reason that the long (weak)- taste memory trace overlaps with the short-trace (CaMKII-GluA1 in the insular cortex) during short-trace timescale.

As mentioned by reviewer 2 ‘they should propose a network of brain areas able to sustain the long-trace different from the network sustaining short-trace’, it is possible that amygdala together with the PFC, or the brain-stem gustatory circuit such as parabrachial nucleus (PBN) and nucleus of solitary tract (NTS) could sustain the long-taste memory trace (Parabuchi and Netser 2014; Carter et al., 2015). These possibilities can be discussed in the updated/revised version of the manuscript.

*There are also several issues about the experimental design, in particular appropriate control experiments. For example, the authors show that MK-801 pre injection obliterated the CS impact, and the authors claim this to be a sign that NMDA receptors are required for CS induced CTA. Unfortunately, the result does not allow to distinguish whether NMDA receptors are required for learning or whether it is required for sensory perception. MK-801 as a general blocker of NMDA receptor function may affect many other aspects of CS learning including perception of taste, general sensory processing. Similar concerns relate to manipulations used in Figure 4 using CNQX.*

We believe that these comments are not valid. It is well demonstrated that NMDAR in the insular cortex is necessary for the associative CTA learning but dispensable for taste perception and incidental taste learning (Rosenblum et al., 1997; Gutierrez et al., 2003; Parkes et al., 2014). Moreover, we replicated these findings and they are included in the manuscript (Figure 3—figure supplement 1).

We have also observed that CNQX micro-injection did not have any effect on the taste perception and incidental taste learning (Figure 4—figure supplement 4).

Reviewer #2:

*[…] In my opinion, the rationale for the study is clearly presented, the experiments are well designed, and the data are appropriately analyzed. I have some major comments meant to encourage the authors to clarify their interpretations of the two parallel memory traces. If these concerns can be addressed the work could yield an important paper.*

We thank the reviewer for the positive comments and encouragement.

*I encourage the authors to elaborate on the existence of the two parallel memory traces. In particular I suggest several points:*

*1) Short-trace: in order to clearly demonstrate that the 1-3 h taste trace is dependent on CaMKII-GluA1 pathway in insular cortex, the authors should evaluate the impact of intra-insular infusion of TatCN21 and CNQX 2h after saccharin consumption and 1h before LiCl (ITI 3h). If their assumption is right these treatments should still impair 3h ITI-CTA.*

We have addressed this constructive comment in the revised manuscript.

As suggested by the reviewer, we have performed two 3h ITI-CTA experiments in which we micro-infused CaMKII inhibitor TatCN21 or AMPAR antagonist CNQX 2h after the taste consumption and administered the LiCl 1h later. Indeed, when we performed this experiment with CNQX, we observed a significant effect of CNQX on 3h ITI-CTA memory (Figure 4). However, we observed a small non-significant trend when we used the TatCN21.

We would like to reiterate three main points: 1) CaMKII-dependent GluA1 induction occurs as early as 1h after the consumption of novel taste. 2) There is a persistent activation of CaMKII for upto 3h after the consumption of a novel taste. 3) The above two new pharmacological experiments, as well as our previous results, demonstrate differential time effects between CaMKII and GluA1. Taken together, we suggest that it is the downstream of CaMKII (i.e. GluA1) that mediates the trace longer than 1h for the association with the US. This interpretation is discussed in the revised manuscript (Results and Discussion).

*2) Long-trace: the authors clearly demonstrate that the long-trace (> 3h ITI) is not dependent on CaMKII-GluA1 pathway in insular cortex. However, as they have previously shown that novel taste is able to enhance protein expression (C/EBPbeta) in insular cortex long after its consumption (i.e. 18h, Merhav et al. 2006), they should consider that the long-trace CTA could still depend on molecular mechanism in insular cortex (but not on CaMKII-GluA1 pathway).*

We agree with the reviewer on this point and presented a detailed explanation in the revised manuscript (Results and Discussion, eighteenth paragraph).

*3) Long-trace: the authors propose that the taste memory trace results in a wider distribution with the passage of time and that there is competition across multiples memory systems. Therefore they should propose a network of brain areas able to sustain the long-trace different from the network sustaining short-trace.*

In the last paragraph of the Results and Discussion, we elaborated on the neuronal network that might underlie long-trace CTA.

[Editors’ note: the authors appealed against the decision and were later encouraged to resubmit.]

Thank you for choosing to send your work entitled "A Molecular Mechanism Underlying Gustatory Memory Trace for an Association in the Insular Cortex" for consideration at eLife. Your submission has now been considered by a Senior editor, the original Reviewing editor, and an additional reviewer, and we are prepared to consider a revised submission with no guarantees of acceptance. The third reviewer's full comments are included below. Before you submit your revised paper please include a thorough explanation of the model and provide additional data.

In addition to the detailed point-by-point response below, we would like to highlight several key additions and changes we have made that we feel would result in a successful resubmission:

As demanded by Reviewer #1 we have expanded our analyses and discussion of the role of NMDAR in the insular cortex (IC) in taste perception (presented in Figure 3—figure supplement 1). We also provide a result (Figure 4—figure supplement 4) that further supports that AMPAR in the IC is dispensable for taste perception. In addition, we provide detailed explanation of the parallel taste memory trace model (Figure 4—figure supplement 5).

As requested by Reviewer #2 we have performed an additional experiments with a 3h inter-time interval of conditioned taste aversion (ITI-CTA) using CaMKII inhibitor TatCN21 (now Figure 3) and AMPA/Kainate receptor antagonist CNQX (now Figure 4).

As requested by Reviewer #3 we have performed an additional standard ITI-CTA experiment with a weak unconditioned stimulus (US) to provide a greater insight into the shift in the associability of the taste memory trace that happens at around 4h after the consumption of a novel taste (Figure 1). In addition, we have conducted two pharmacological experiments in which we micro-infused an NMDAR antagonist into the IC 30min or 4h after taste consumption in a 1h ITI-CTA with a weak US or a 5h ITI-CTA with a strong US, respectively (Figure 3—figure supplement 3).

At the request of all the reviewers we have substantially expanded our discussion concerning the molecular and circuit mechanisms that might underlie long-trace memory and parallel memory trace model (Figure 4—figure supplement 5).

Reviewer #1:

*A major claim of the authors is that the drop in learning efficacy after 3 hours signifies the presence of two parallel mechanisms of CTA acquisition, a claim that is also supported by a previous study from the same group (Merhav & Kornblum, 2008) that shows that protein synthesis inhibitors have a stronger impact on CTA shorter than 3 hours. However, this claim receives overall weak support, as retention of the CS signal may simply be decaying nonlinear with time; or alternatively, the underlying processes linked to novel taste acquisition and induction of CTA crosses below a threshold for efficient learning acquisition after app. 3 hours.*

We would like to expand upon our previous responses to Reviewer #1. We would like to emphasize that three different lines of evidence (behavioural, Figure 1; biochemical, Figure 2 and Figure 4; and pharmacological, Figure 3 and Figure 4) support the possibility of two parallel taste memory traces.

The reviewer commented that ‘retention of the CS signal may simply be decaying nonlinear with time’. It is clearly a possibility; however, our experiments and the respective statistical analyses did not support this possibility, but instead support the parallel memory traces hypothesis.

First, non-parametric one way ANOVA (Kruskal-Wallis) and non-parametric repeated measures ANOVA (Friedman’s) revealed that the aversion index measure following 1h, 2h, and 3h ITI-CTAs significantly differs from that of 4h-8h ITI-CTAs (Figure 1, and [Supplementary-material SD1-data]). The follow-up unbiased cluster analysis revealed that 1h-8h ITI-CTAs fall into two different clusters; 1h,2h, and 3h ITI-CTAs as small cluster, and 4h-8h ITI-CTAs as large cluster and they are statistically different from each other (Mann Whitney U test; [Supplementary-material SD1-data]). Moreover, as we have presented in the revised manuscript (Figure 1), the associability of the CS with the weak US completely drops at around 4h-5h.

Secondly, in striking opposite to the reviewer’s comment, in Figure 3—figure supplement 4 we have demonstrated (consistent with the previous reports, Kalat and Rozin, 1973; Gutierrez et al., 2003) that the retention of the CS signal gets stronger with time.

The reviewer again presented an alternative comment ‘the underlying processes linked to novel taste acquisition and induction of CTA crosses below a threshold for efficient learning acquisition after app. 3 hours’. This is counter-intuitive to what we have presented in the manuscript: that the underlying process (NMDAR-CaMKII-GluA1 pathway) for formation of CTA with upto a 3h ITI is different from those who have an ITI >3h (4h-8h ITI CTA).

*The authors state that short-memory trace is required to generate the weak memory trace which lasts longer, yet the authors claim that the two memory traces are parallel. This is confusing, as when one trace requires the other one, they are by definition not parallel processes.*

We see the point of the reviewer. We would like to clarify this point with the following graphical representation (please see Figure 4—figure supplement 5).

*There are also several issues about the experimental design, in particular appropriate control experiments. For example, the authors show that MK-801 pre injection obliterated the CS impact, and the authors claim this to be a sign that NMDA receptors are required for CS induced CTA. Unfortunately, the result does not allow to distinguish whether NMDA receptors are required for learning or whether it is required for sensory perception. MK-801 as a general blocker of NMDA receptor function may affect many other aspects of CS learning including perception of taste, general sensory processing. Similar concerns relate to manipulations used in Figure 4 using CNQX.*

Reviewer #2:

Please see the previous responses to Reviewer #2.

Reviewer #3:

*[…] I have several concerns regarding the conclusions of the authors about the two parallel memory traces, which should be addressed before the article can be accepted for publication. 1) The authors should at the very least hypothesize regarding the possible molecular mechanisms that might underlie the long but weak memory trace formation (e.g. NMDAR dependent), or even better test their hypothesis by applying the relevant pharmacological intervention.*

In order to directly address the reviewer’s concerns regarding the possible molecular mechanisms (NMDAR dependent) that might underlie the long but weak memory trace formation, we conducted an additional experiment in which we injected NMDA receptor antagonist APV into the insular cortex 4h after novel taste consumption in a 5h ITI-CTA experiment and found no effect on CTA memory (Figure 3—figure supplement 3). The result of this experiment indicates that long-trace taste memory is not mediated by the NMDA receptor in the IC.

We strongly feel that the data we provide in the manuscript allows us to confidently argue that the NMDA receptor dependent CaMKII-GluA1 signaling in the insular cortex mediates the short-trace CTA memory formation. Moreover, recently we have reported that the NMDAR dependent function of the hippocampus in mice is crucial when there is a long temporal gap between taste and malaise (Chinnakkaruppan et al., 2014). And also, a previous study reported that a hippocampus lesion strongly interferes with the long- but not short-trace CTA in rats (Koh et al., 2009).

Taken together, we think that the dominance of the gustatory trace in the IC weakens with time while other components in CTA learning and the respective brain circuit (for instance, the necessity of NMDA receptors in the hippocampal dentate gyrus) may play a more dominant role with time.

As also suggested by the Reviewer #2, we have presented possible molecular and circuit mechanisms that might underlie the long-trace CTA in the Discussion part of the revised manuscript (last paragraph).

*2) Related to this issue, it would be important to examine the effect of intra-insular infusion of MK-801 at a time interval longer than 3 hours to examine the possible effect of NMDAR on the weak memory trace.*

Please refer our comment above.

*3) Although the authors nicely demonstrate the temporal effects of the CTA (CS + US) acquisition traces, there is no quantification of the impact of the novel taste (CS) and malaise (US) on the formation of these memory traces. The authors could apply a weak CTA learning by reducing the LiCl concentration to address this question. Alternatively, they could induce weak learning by using latent inhibition, where the CS is familiarized before association. They should test: (a) whether this weaker learning can be acquired when association is made during the short (1-3 hrs) interval and long (4-8 hrs) interval and (b) is it sensitive to the CaMKII-GluA1 proposed mechanism or perhaps to a NMDAR dependent one.*

These comments are very interesting and we have addressed them in the revised manuscript.

First, we performed an ITI-CTA experiment with the weaker US (reduced LiCl concentration; 0.025M LiCl compared to strong 0.15M LiCl, 2% b.w.) and found that the CTA was formed following the interval pertaining to the short-trace, and interestingly, also following a 4h but not 5h interval, suggesting that there is a shift in the associability at around 4h- 5h after exposure to the taste stimulus. We have included this result in the revised Figure 1 and discussed it in the text accordingly (Results and Discussion, second paragraph).

Second, as shown previously with the strong US (Rosenblum et al., 1997), we found an attenuated CTA when the NMDAR antagonist, APV, was micro-injected 25min after novel taste consumption in a 1h ITI-CTA experiment with the weaker US (Revised Figure 3—figure supplement 3). This result, together with a result presented in our response to major comment 1, suggests the dependency of the short-trace but not long-trace CTA on NMDAR in the IC.